# SIRT3 promotes auditory function in young adult FVB/nJ mice but is dispensable for hearing recovery after noise exposure

Sally Patel[1], Lisa Shah[1], Natalie Dang[1], Xiaodong Tan[2], Anthony Almudevar[3], Patricia M. White[1]*

1 Department of Neuroscience, Ernest J. Del Monte Institute for Neuroscience, University of Rochester School of Medicine and Dentistry, Rochester, New York, United States of America, 2 Department of Otolaryngology-Head and Neck Surgery, Feinberg School of Medicine, Northwestern University, Chicago, Illinois, United States of America, 3 Department of Biostatistics and Computational Biology, University of Rochester School of Medicine and Dentistry, Rochester, New York, United States of America

* patricia_white@urmc.rochester.edu

**Data Availability Statement:** All ABR, DPOAE, and confocal imaging files generated for this project are available through UR Research (http://hdl.handle.net/1802/35758).

## Abstract

Noise-induced hearing loss (NIHL) affects millions of people worldwide and presents a large social and personal burden. Pharmacological activation of SIRT3, a regulator of the mitochondrial oxidative stress response, has a protective effect on hearing thresholds after traumatic noise damage in mice. In contrast, the role of endogenously activated SIRT3 in hearing recovery has not been established. Here we tested the hypothesis that SIRT3 is required in mice for recovery of auditory thresholds after a noise exposure that confers a temporary threshold shift (TTS). SIRT3-specific immunoreactivity is present in outer hair cells, around the post-synaptic regions of inner hair cells, and faintly within inner hair cells. Prior to noise exposure, homozygous *Sirt3-KO* mice have slightly but significantly higher thresholds than their wild-type littermates measured by the auditory brainstem response (ABR), but not by distortion product otoacoustic emissions (DPOAE). Moreover, homozygous *Sirt3-KO* mice display a significant reduction in the progression of their peak 1 amplitude at higher frequencies prior to noise exposure. After exposure to a single sub-traumatic noise dose that does not permanently reduce cochlear function, compromise cell survival, or damage synaptic structures in wild-type mice, there was no difference in hearing function between the two genotypes, measured by ABR and DPOAE. The numbers of hair cells and auditory synapses were similar in both genotypes before and after noise exposure. These loss-of-function studies complement previously published gain-of-function studies and help refine our understanding of SIRT3's role in cochlear homeostasis under different damage paradigms. They suggest that SIRT3 may promote spiral ganglion neuron function. They imply that cellular mechanisms of homeostasis, in addition to the mitochondrial oxidative stress response, act to restore hearing after TTS. Finally, we present a novel application of a biomedical statistical analysis for identifying changes between peak 1 amplitude progressions in ABR waveforms after damage.

**Funding:** PMW, R01 DC14261, National Institute for Deafness and Communication Disorders, https://www.nidcd.nih.gov/ XT, American Hearing Research Foundation, https://www.american-hearing.org/ The funders had no role in study design, data collection and analysis, decision to publish, or preparation of the manuscript.

**Competing interests:** The authors have declared that no competing interests exist.

## Introduction

Noise-induced hearing loss (NIHL) affects at least 10 million adults in the United States [1], including over a million veterans [2]. Noise-induced auditory dysfunction, including tinnitus and NIHL, is the most common disability among former combat soldiers, costing the Veteran's Administration over a billion dollars annually [2]. NIHL is a form of acquired hearing loss, which can be associated with greater levels of anxiety [3], emotional distress [4], and perceived stigmatization [5], as well as poorer health outcomes [6]. There is no approved biological treatment for NIHL [1], and it can only be prevented by physically avoiding noise exposure.

Recent progress has been made in identifying genetic variants that predispose individuals to NIHL in occupational settings [7, 8]. Gene variants encoding proteins that modulate oxidative stress are over-represented in these studies (for review, see [9]). More specifically, gene variants that reduce mitochondrial function enhance susceptibility to acquired hearing loss from noise [10, 11], ototoxic drugs [12], and age-related hearing loss [13, 14]. These facts support a model where proteins promoting mitochondrial function and counteracting oxidative stress protect hearing from excessive noise. SIRT3 is a mitochondrial lysine deacetylase [15] that promotes an effective oxidative stress response [16] from mitochondrial enzymes, including Superoxide Dismutase 2 (SOD2, [17]). SIRT3 activation through dietary restriction has previously been shown to protect cochlear outer hair cells (OHCs) from aging. The positive effect was only observed in wild-type, not homozygous *Sirt3-KO* mice [18]. Genetic or pharmaceutical activation of sirtuins was also shown to rescue hearing from traumatic noise damage in a SIRT3-dependent manner [19]. These findings indicate that exogenous activation of SIRT3 is sufficient to protect cochlear cells from damaging insults. The same study also showed that both homozygous Sirt3-KO mice and wild-type controls had similar levels of damage from a permanent threshold shift (PTS)-inducing noise exposure in the absence of exogenous SIRT3 activation [19]. However, whether SIRT3 has a role in protecting the cochlea from a sub-traumatic noise exposure has not been established.

Here we present studies investigating whether there is a requirement for SIRT3 in the recovery of hearing thresholds in adult mice after a temporary threshold shift (TTS). Since TTS is associated with a recoverable disrupted cellular process rather than hair cell loss [20], SIRT3's activity in hearing recovery from sub-traumatic noise can be determined by comparing wild-type and *Sirt3-KO* mice. These loss-of-function studies complement the already-published gain-of-function studies. Multiple labs have successfully used this method to identify genes [21] and conditions [22] that modulate susceptibility to noise damage. Noise exposure can eliminate high-frequency auditory synapses [20, 23] in a glutamine-dependent manner [24], and the restoration of auditory synapses was proposed to be the mechanism by which SIRT3 reduced NIHL [19]. To evaluate any requirement for endogenous SIRT3 activity in resilience from noise damage, we employed a noise exposure that is 80% of the energy level needed to induce synaptopathy [20, 23]. This treatment induces small or negligible permanent ABR threshold shifts [21], and does not cause OHC death [21]. By exposing homozygous *Sirt3-KO* mice and their wild-type littermates to sub-traumatic noise, we sought to uncover Sirt3's role in preserving these cellular structures and functions after TTS.

## Materials and methods

### Animal usage

All experiments were performed in compliance with the US Department of Health and Human Services, and were approved by the University Committee on Animal Resources at the

University of Rochester Medical Center (#2010–011, PI Patricia White) or the IACUC at Northwestern University. Male *Sirt3-KO* mice (129-Sirt3tm1.1Fwa/J; stock number 012755; The Jackson Laboratory) were bred four times to FVB/nJ females (stock number 001800, The Jackson Laboratory). In other experiments, we have found that four generations is sufficient to confer youthful noise damage sensitivity similar to that of congenic FVB/nJ [21, 23]. Heterozygotes with different parents were bred together to obtain both knockout and wild-type littermates. Both males and females were used. In the first five litters generated, fourteen homozygous *Sirt3-KO* and thirteen wild-type littermates were generated and then exposed to noise. Four additional litters were generated to produce homozygous *Sirt3-KO* mice and wild-type littermates, which were not exposed to noise, for histology. Mice were given ample nesting materials and small houses within their home cage. Mice were euthanized with gradual $CO_2$ exposure in their home cage. One minute after the cessation of breathing, mice were decapitated to ensure death.

For genotyping, DNA was obtained from 2-mm tail samples from P21-P25 weanlings using sterilized scissors without anesthesia, in accordance with University guidelines. Samples were digested overnight in Proteinase K (IBI Sciences) solution at 65˚C followed by phenol/chloroform extraction. The KAPA Taq PCR kit (Sigma, BK1000) was used in conjunction with three primer sequences (wild type, `CTT CTG CGG CTC TAT ACA CAG`; common, `TGC AAC AAG GCT TTA TCT TCC`; mutant, `TAC TGA ATA TCA GTG GGA ACG`) to identify genotypes.

## Noise exposure

Awake two month old mice were exposed to noise limited to the 8–16 kHz octave band at 105 decibels for 30 minutes. Mice were each placed into individual triangular wire mesh cages, 12 cm x 5 cm x 5 cm, in an asymmetric plywood box with a JBL2250HJ compression speaker and JBL2382A biradial horn mounted on the top. This apparatus was contained within a sound booth. The speaker was driven by a TDT RX6 multifunction processor and dedicated attenuator, and controlled with TDT RPvdsEx sound processing software. The sound level was calibrated with a Qwest sound meter, and the sound level was checked each morning with a calibrated iPhone using the FaberAcoustical SoundMeter app. Mice were exposed to noise between 9 am and 4 pm. No analgesia was provided; however, after noise exposure all mice were ambulatory and did not show common signs of distress, such as failure to groom, hunched posture, or lack of movement. Mice were observed daily for three days after the noise exposure procedure for signs of fighting. Fighting mice were isolated from their cage mates in separate housing.

## Auditory testing

Mice were tested at 7 weeks of age (pre-test), one day after receiving noise exposure, and again fourteen days later. Mice were exposed to noise at P60. Auditory testing was conducted using a Smart EP Universal Smart Box (Intelligent Hearing Systems) with high-frequency speakers from Tucker Davis Systems. Mice were anesthetized with an intraperitoneal injection of ketamine (80 mg/kg) in a sterile acepromazine/saline mixture (3 mg/kg). A 10B+ (high frequency transducer/stimulator) probe was placed at the opening to the external auditory meatus. Sound production in this system was calibrated with a Qwest sound meter every six months.

Auditory brainstem response (ABR) stimuli were 5-ms clicks, or 5-ms tone pips presented at 5 frequencies between 8 and 32 kHz. Stimuli began at 75 dB amplitude and decreased by 5 dB steps to 15–25 dB. 512 sweeps were averaged for each frequency and amplitude. Electrical responses were measured with three subdermal needle electrodes (Grass): one inserted

beneath each pinna, and a third, the ground electrode, placed at the vertex. ABR thresholds for a particular frequency were determined by any part of the last visible trace (dB). The person scoring the waveforms was blinded to genotype and time point. Masking and randomization was performed by drawing cards from a fair deck and renaming files appropriately.

For distortion product otoacoustic emissions (DPOAE), we measured the amplitude of evoked otoacoustic emissions to paired pure tones of frequencies f1 and f2, where f1/f2 = 1.2 and the f1 level was 10 dB above f2. Thirty-two sweeps were made in 5 dB steps starting with f1 at 20 dB and ending at 65 dB. The DPOAE threshold was calculated by interpolating the f2 level that would produce a 3 dB response.

## Antibodies

The following primary antibodies were used: mouse IgG1 anti-SIRT3 antibody (1:200; Novus Biologicals; RRID:AB_2818991); goat anti-Oncomodulin antibody (OCM; 1:1000; Santa Cruz; RRID:AB_2267583), rabbit anti-Myosin7a (MYO7; 1:200; Proteus; RRID:AB_10013626) mouse anti-CTBP2 (aka C-Terminal Binding Protein 2; 1:200; BD Transduction Laboratories; RRID:AB_399431), and mouse anti-GRIA2 (aka GluR2/GluA2; 1:2000; Millipore; RRID: AB_2113875). The following secondary antibodies were purchased from Jackson Immuno Research: Donkey Anti-Mouse AF488 (1:500; RRID:AB_2340849), Donkey Anti-Rabbit AF594 (1:500; RRID:AB_2340622), Donkey Anti-Rabbit AF647 (1:200; RRID:AB_2340625), Donkey Anti-Goat AF647 (1:200; RRID:AB_2340438), Goat Anti-Mouse AF594 (IgG1, 1:500; RRID:AB_2338885), AF488 Goat Anti-Mouse (IgG2a, 1:500; RRID:AB_2338855). For the images in Fig 1, an AF568 Goat Anti-Mouse (IgG1, 1:200, Thermo Fisher, RRID: AB_2535766) was used.

## Tissue preparation for immunostaining

Cochlear organs were dissected out of freshly euthanized animals. Their stapes were removed, and a hole was made in their apical tips to allow for adequate fluid exchange. Tissues were immersed in 4% paraformaldehyde (PFA) in PBS for at least overnight, and decalcified in 0.1M EDTA at 4˚C on a rotating platform for four days. These cochleae were decalcified in 10% EDTA in 1X PBS (diluted from 10X PBS, Invitrogen) for 2–3 days. All whole mount preparations were microdissected into turns as previously described [25, 26], and the tectorial membrane was carefully removed. Cochleae were mapped using the ImageJ plug-in from Massachusetts Eye and Ear Infirmary.

## Immunostaining

For anti-SIRT3 staining, cochlear pieces were first soaked in 30% sucrose on a shaker for 20 minutes, and then transferred to dry ice for 10–15 minutes or until the sucrose was completely frozen. The samples were allowed to thaw at room temperature and washed with 1X PBS 3 times, rinsing for 20 minutes in between on a shaker. After blocking with 5% serum + 1% Triton, the primary antibody mix diluted in block was added and incubated overnight at 37˚C. After rinsing 3 times in PBS (10 minutes each), secondary antibody mix (diluted in block and including Hoechst or DAPI at 1:1000) was added and incubated away from light for 2 hours at 37˚C. The pieces were then washed another 3 times with 1X PBS and mounted on slides.

For post-injury histological characterization, dissected mapped turns were immersed in 30% sucrose, flash frozen in liquid nitrogen, allowed to thaw, washed in room temperature Dulbecco's PBS (Gibco), and blocked for one hour in 1% Triton / 5% donkey serum in PBS. Primary antibody incubations of anti-MYO7, anti-OCM, anti-CTBP2, and anti-GRIA2 were performed at 37˚C for 20 hours. The tissue was washed in PBS, and secondary antibody

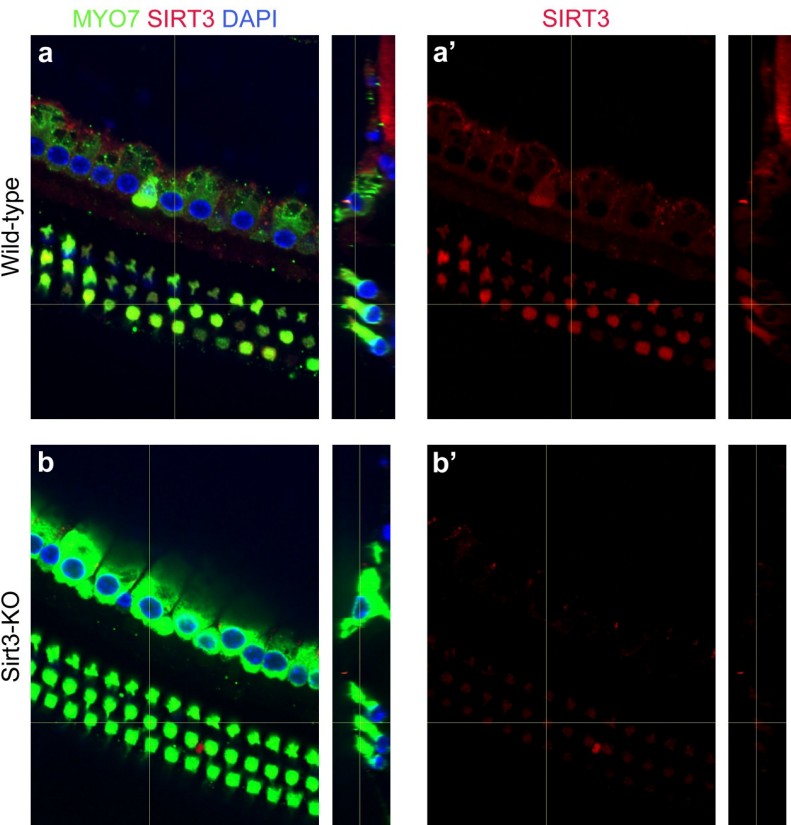

**Fig 1. SIRT3 is expressed in sensory cells of the adult mouse cochlea.** a. Cochlear whole mount preparation from a wild-type adult mouse, immunolabeled with antibodies specific to MYO7 (green) and SIRT3 (red), and co-stained with DAPI (blue) to reveal nuclei. The vertical yellow lines show the positions from which the side views are taken. (a') shows anti-SIRT3 immunoreactivity only. b. Cochlear whole mount preparation from a *Sirt3-KO* adult mouse, identically labeled and imaged. This control was used to set the background to reveal SIRT3-specific staining.

incubation was performed at 37°C for an additional 2 hours, with both MYO7 and OCM labeled with 647-conjugated antibodies. This whole mount protocol was kindly provided by Leslie Liberman. All tissue was mounted using ProLong Gold (Fisher). Whole mounts were placed between two 50 mm coverslips for imaging.

## Confocal microscopy and image processing for figures

Anti-SIRT3 immunostaining was analyzed at the Center for Advanced Microscopy/Nikon Imaging Center at Northwestern University, using a Nikon A1 Laser Scanning or Nikon W1 Dual CAM Spinning Disk imaging setup. Appropriate excitation and emission settings were used and fixed for all panels in the same images presented in the paper. Post-injury histological characterizations were done on an Olympus FV1000 laser scanning confocal microscope at the Center for Advanced Light Microscopy and Nanoscopy at URMC. ImageJ (NIH) was used to Z-project maximal brightness in confocal stacks. Photoshop (Adobe) was used to set maximal and background levels of projections for the construction of figures.

For cochleogram analysis, composite images for cochleograms were assembled in Photoshop by pasting each optical section into its own layer, and merging the pieces of the optical sections where hair cells were evident. Alternatively, projections of confocal stacks were used when individual hair cells could be clearly distinguished. Composite or projected images for

the mapped regions of each organ were assembled in a single file in Photoshop. 100 micron lengths were pasted onto the images on the row of pillar cells, and OHCs were counted to determine the percent lost. Where the loss of OHCs was great enough (>30%), an average OHC count, determined from low-frequency regions of the same cochlea, was used as the denominator [27].

Synaptic components of inner hair cells (IHCs) were imaged with a 100X oil lens at 2X magnification. To obtain 3D images of IHCs, the confocal files were opened in Amira 6.0 (Visualization Science Group), appropriately thresholded, and rotated to better display the IHCs. To count ribbon synapses, we visually quantified red/green synaptic puncta, as indicated by colocalized staining within IHCs on projected confocal stacks. The analyst was blinded to genotype, frequency and condition, and the images were randomized in presentation, by renaming the images according to cards drawn from a fair deck. Synapse numbers were then divided by the number of hair cells in the field to obtain a biological replicate. In previous publications, we have used automated synapse detection methods; however, in some of these preparations the staining background for GRIA2 was too variable for computational approaches that use thresholding. This was determined by comparing the number of synapses obtained through visual inspection to those obtained through automated means [28].

### Statistical analysis

Statistical tests for ABR and DPOAE thresholds, and for DPOAE input-output functions, were performed in R using standard functions. The data distribution for cochleograms did not meet the Shapiro test for normality, and was therefore assessed with the Kruskal-Wallis rank sum test. Both functions were also performed in R.

To calculate significance levels for amplitude progressions, we compared the integrals of the amplitude progressions, as they reflect the total number of coordinately signaling spiral ganglion neurons. To model the progressions, generalized estimating equations (GEE) were used. These are suitable for the analysis of longitudinal data when population level mean responses suffice to resolve hypotheses. The correlation among responses induced by the repeated measure design were thus modelled by constructing a suitable covariance matrix [29].

Generalized linear model conventions were used. Responses were modeled as Poisson variates, with varying scale parameter to account for overdispersion. The exchangeable correlation model was used to account for within-sample correlation. The link function was $g(\mu) = log(\mu)$ so that $\mu = e^{\eta}$, where $\eta$ is a linear prediction term dependent on threshold x and binary group factor $I$. The null hypothesis of no group effect was modelled as $H_0: \eta = poly(x, d)$, and the alternative hypothesis was modelled as H$\alpha$: $\eta = poly(x, d) * I$, where $poly(x, d)$ is an order $d$ polynomial with unknown parameters. For the baseline, 14 day post noise (DPN) and KO analyses, $d = 2$ was used, while for the WT analysis $d = 4$ proved to be a significantly better fit. Observed significance levels were obtained both by the $\chi^2$ approximation method of **geeglm**, as well as by a parametric bootstrap method. P-values obtained by both methods were comparable. However, the bootstrap procedure was the more conservative, so these P-values were used (see Table 1).

### Results

Immunofluorescence for SIRT3 protein reveals its expression pattern in the adult mouse cochlea (Fig 1). Inner and outer hair cells (IHC & OHC) were visualized with anti-Myosin7 antibodies (Fig 1, MYO7, green), and nuclei were revealed with DAPI (Fig 1, blue). SIRT3 immunoreactivity was evident in OHCs and especially in the positions of nerve root endings

**Table 1. Integrals and p-values for 32 kHz ABR voltage progressions for homozygous *Sirt3-KO* and wild-type mice.**

| Fixed Factor | | Integral | | Integral | Integral Difference | P-values | |
| --- | --- | --- | --- | --- | --- | --- | --- |
| | | | | | | Integral Diff | Curve Shape |
| Pre-test | WT | 11645 | KO | 6577 | 5067 | 0.046 * | 0.008 |
| 14DPN | WT | 6185 | KO | 6534 | -349 | 0.874 | 0.028 |
| KO | Pre-test | 6552 | 14DPN | 6702 | -150 | 0.914 | 0.001 |
| WT | Pre-test | 11046 | 14DPN | 5747 | 5299 | 0.002 *** | 0.138 |

and supporting cells surrounding the IHCs (Fig 1a, red). The latter finding is readily observed in the side view (Fig 1a, cf. red to green). SIRT3 immunoreactivity was significantly greater than that of age-matched, exposure-matched preparations from homozygous *Sirt3-KO* mice (Fig 1b, red), demonstrating antibody specificity. Endogenous SIRT3 protein in the cochlea was not sufficiently abundant to visualize on a western blot. Messenger RNA levels for sirtuins have been previously described in the adult mouse cochlea, with *Sirt3* mRNA significantly expressed in adult mouse IHC and OHC [30].

Hearing thresholds for homozygous *Sirt3-KO* and wild-type littermates were evaluated by auditory brainstem response (ABR) and by distortion product otoacoustic emissions (DPOAE). While others have reported no difference in auditory thresholds between young adult homozygous *Sirt3-KO* and wild-type littermates [18, 19, 31], those experiments were performed using mice of the C57BL/6J strain. To assess the response to noise damage, we bred Sirt3-KO mice to FVB/nJ mice, which have good hearing and a youthful susceptibility to noise damage [32]. Littermates of both genotypes underwent hearing tests at around P50. On FVB/nJ, homozygous Sirt3-KO mice demonstrate slightly but significantly higher ABR thresholds on average compared to wild-type littermates (Fig 2b, pink traces, p = 0.004 for genotype, two-way ANOVA, n = 27 mice total, noted as **). However, no single frequency was significantly worse (p = 0.053, 0.39, 0.41, 0.53, 0.059 for 8, 12, 16, 24, and 32 kHz frequencies respectively, two-tailed pairwise t-tests at each frequency with Bonferroni correction, n = 27 mice total). DPOAE threshold averages were identical for both genotypes (Fig 2e, p = 0.43 for genotype, two-way ANOVA, n = 27 mice total, noted as ns).

To evaluate SIRT3's role in hearing recovery, we exposed mice to a noise dose that confers little permanent damage to awake, two month old, wild-type FVB/nJ mice [21, 23]. An exposure to an 8–16 kHz octave band at 105 dB for thirty minutes is sufficient to elevate auditory brainstem response (ABR) thresholds in mice one day after noise exposure, which is fully recovered in two weeks. This exposure reduces high-frequency peak 1 amplitudes two weeks after exposure, but does not significantly eliminate high-frequency synapses [23]. Mice previously tested with ABR and DPOAE were exposed to noise at P60 and received two additional hearing tests at P61 and P74 (Fig 2a). At P74, mice were euthanized for histological analysis.

Noise exposure significantly increased ABR thresholds in both homozygous Sirt3-KO mice (Fig 2b cf. c, pink traces, p = 6x10$^{-14}$, for time point, two way ANOVA, n = 14 mice) and their wild-type littermates (Fig 2b cf. c, black traces, p = 2x10$^{-16}$ for time point, two-way ANOVA, n = 13 mice). In contrast to the comparison of ABR thresholds prior to noise exposure, thresholds for the two genotypes were not different at 1 DPN (Fig 2c, p = 0.35 for genotype at 1 DPN, two-way ANOVA, n = 27 mice). At 14 DPN, there was also no difference in ABR thresholds between genotypes (Fig 2d, p = 0.85 for genotype at 14 DPN, two-way ANOVA, n = 27). We similarly calculated thresholds for distortion product otoacoustic emissions (DPOAE) for the same mice at the same time points (Fig 2f and 2g). DPOAE thresholds for both genotypes were similarly elevated at 1 DPN for both genotypes, and recovered to the same levels at 14 DPN (p = 0.13 for genotype, multivariate ANOVA, n = 27 mice).

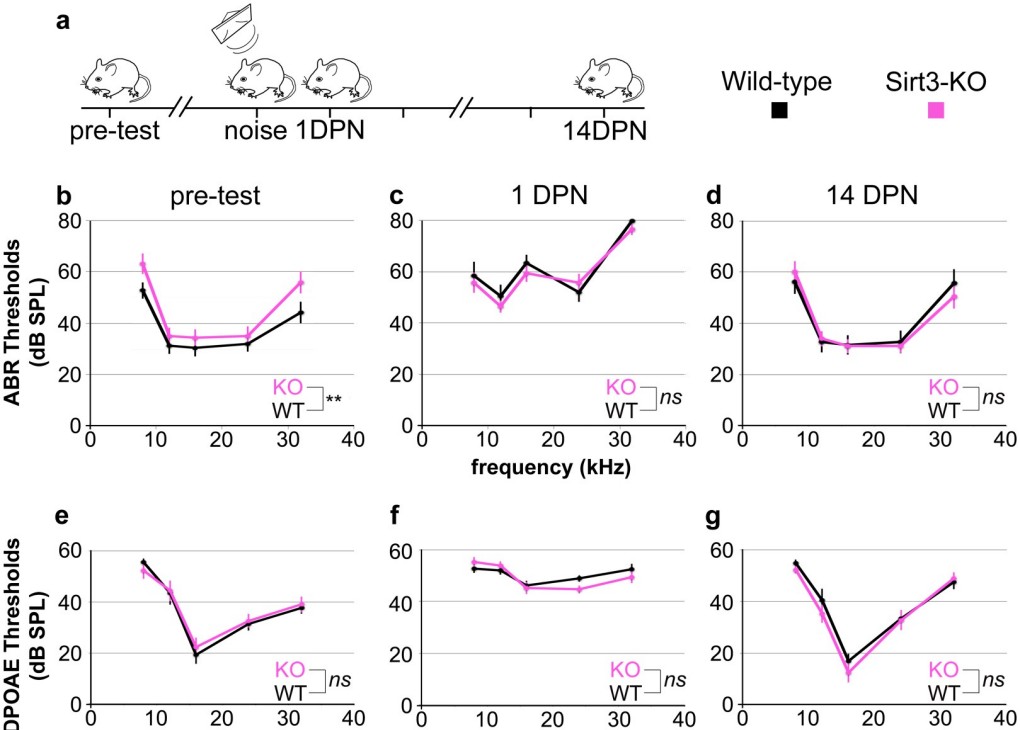

**Fig 2. SIRT3 loss of function impacts initial auditory thresholds but not hearing recovery from TTS.** a. The time course of experiments is shown. Homozygous *Sirt3-KO* mice and wild-type littermates are tested for ABR and DPOAE thresholds prior to noise exposure (pre-test). At P60, they are exposed to an 8–16 octave band noise at 105 dB for 30 minutes (noise), which is sufficient to drive temporary threshold shifts in this strain [21, 23]. Their hearing thresholds are assessed again at 1 day post noise (1 DPN) and 14 days post noise (14 DPN). ABR and DPOAE are assessed at 8, 12, 16, 24, and 32 kHz for all panels. b. Mean ABR thresholds prior to noise exposure for homozygous *Sirt3-KO* mice (n = 14, pink) and wild-type littermates (n = 13, black). Mice were approximately P50 when tested. Homozygous *Sirt3-KO* mice had significantly worse overall hearing (p = 0.004 for genotype, two way ANOVA, n = 27 mice total, noted as \*\*); however, no single frequency was significantly worse (p = 0.053, 0.39, 0.41, 0.53, 0.059 for the five frequencies respectively, two-tailed pairwise t-tests at each frequency with Bonferroni correction, n = 27 mice total). c. Mean ABR thresholds at 1 DPN for homozygous *Sirt3-KO* mice (n = 14, pink) and wild-type littermates (n = 13, black). No significant difference was seen between genotypes (p = 0.35 for genotype, two-way ANOVA, n = 27 mice total, noted as n.s.). d. Mean ABR thresholds at 14 DPN for homozygous *Sirt3-KO* mice (n = 14, pink) and wild-type littermates (n = 13, black). No significant difference was seen between genotypes (p = 0.85 for genotype, two way ANOVA, n = 27 mice total). Overall ANOVA for all three time points showed no significant difference between genotypes (p = 0.33, multi-way ANOVA, n = 27 mice total, noted as n.s.). e. Mean DPOAE thresholds prior to noise exposure for homozygous *Sirt3-KO* mice (n = 14, pink) and wild-type littermates (n = 13, black). No significant difference was seen between genotypes (p = 0.43 for genotype, two-way ANOVA, n = 27 mice total, noted as n.s.). f. Mean DPOAE thresholds at 1 DPN for homozygous *Sirt3-KO* mice (n = 14, pink) and wild-type littermates (n = 13, black). No significant difference was seen between genotypes (p = 0.50 for genotype, two-way ANOVA, n = 27 mice total, noted as n.s.). g. Mean DPOAE thresholds at 14 DPN for homozygous *Sirt3-KO* mice (n = 14, pink) and wild-type littermates (n = 13, black). No significant difference was seen between genotypes (p = 0.25 for genotype, two-way ANOVA, n = 27 mice total, noted as n.s.). Error bars: s.e.m.

We evaluated the requirement for SIRT3 on ABR peak 1 amplitudes of homozygous *Sirt3-KO* mice and their wild-type littermates before and after noise damage. Mean ABR thresholds for homozygous *Sirt3-KO* mice were 55.7 dB ± 4.1 at 32 kHz, compared to 44.2 ± 4.1 for wild-type littermates, which was not different when adjusted for multiple comparisons. Note that thresholds are determined by the continued presence of any part of the waveform, and not solely by the presence of peak 1. To assess differences between peak 1 amplitude progressions, we used generalized estimating equations (GEE) to build linear models of the correlated data (see Methods), as these are suited for the analysis of longitudinal data

when population level mean responses suffice to resolve hypotheses [29]. We then compared the integrals of these models between genotypes and conditions. We propose that the amplitude integrals (Table 1) better represent the increased recruitment of auditory synapses as the stimulus is amplified, compared to using the amplitude values themselves.

Fig 3 shows pairwise comparisons of data obtained from each mouse (Fig 3a–3d, light lines). The linear models obtained through GEE analysis are displayed on the same graphs (Fig 3a–3d, heavy lines). Latency values for peak 1 at both time points are also shown (Fig 3e and 3f). The values of the amplitude integrals and their associated p-values are listed in Table 1 (see Methods for a full description). Mean peak 1 amplitude integrals were significantly reduced for homozygous *Sirt3-KO* mice compared to wild-types at 32 kHz, after adjusting for multiple comparisons using this approach (p = 0.046, parametric bootstrap method, n = 27 mice, indicated with *). At 14 DPN, mean peak 1 amplitude integrals at 32 kHz were significantly reduced in wild-type littermates compared to measurements before noise exposure (p = 0.002, parametric bootstrap method, n = 13 mice, indicated with **). At 14 DPN, mean peak 1 amplitude integrals were similar when the genotypes were compared (p = 0.87, parametric bootstrap method, n = 27 mice, indicated with *ns*). Mean peak 1 amplitude integrals at 32 kHz in homozygous *Sirt3-KO* mice were unchanged before and 14 days after noise (p = 0.914, parametric bootstrap method, n = 14, indicated with *ns*). Latencies for peak 1 were not different between genotypes at either time point (Fig 3e and 3f, cf. black to red, Kruskal-Wallis rank sum test, n = 27 mice, p = 0.68 for pre-tests and p = 0.26 for 14 DPN, indicated with *ns*).

We evaluated the requirement for SIRT3 on DPOAE fine structure by comparing input-output functions for homozygous *Sirt3-KO* and wild-type mice, prior to noise exposure and at 14 DPN (Fig 4). Values for each L2 frequency are displayed separately, comparing homozygous *Sirt3-KO* mice (Fig 4, pink) to wild-type littermates (Fig 4, black). The curves overlapped at each frequency, and no significant differences were observed (p = 0.07 for genotype, multivariate ANOVA, n = 27 mice). Taken together, these data suggest that the absence of endogenous SIRT3 activity does not confer susceptibility to permanent functional impairment from noise exposure, by these measures.

We sought to determine if OHCs or IHCs are more susceptible to cell death from noise in the absence of SIRT3. We prepared mapped cochlear organs for whole mount imaging of OHCs and IHCs for cochleogram analysis. The cochleae were divided into 100 micron segments along the line of pillar cells, surviving OHCs and IHCs were quantified for each segment, and the percent of lost hair cells were plotted as a function of distance along the cochlear length (Fig 5). Basal OHCs were more prone to losses in both the wild-type and homozygous *Sirt3-KO* mouse in the absence of noise damage (Fig 5, cf. b to h), with some variability between biological replicates (Fig 5b, cf. purple and green). The TTS noise exposure did not potentiate OHC or IHC losses in wild-type littermates (Fig 5e cf. b, p = 0.18 (OHCs), f cf. c, p = 0.80 (IHCs), Kruskal-Wallis rank sum test, n = 3,4 biological replicates). Cell losses were not different between homozygous Sirt3-KO mice and their wild-type littermates in the absence of noise (Fig 5h cf. b p = 0.71 (OHCs), i cf. c, p = 0.45 (IHCs), Kruskal-Wallis rank sum test, n = 4 biological replicates). Like wild-type littermates, the TTS noise exposure did not potentiate OHC or IHC losses in the homozygous *Sirt3-KO* mouse (Fig 5k cf. h, p = 0.49 (OHC), l cf. i, p = 0.76 (IHC), Kruskal-Wallis rank sum test, n = 4 biological replicates). These data indicate that SIRT3 loss of function does not significantly impact hair cell survival after mild noise insult.

SIRT3 immunoreactivity is also present in cellular structures immediately adjacent to IHCs, around regions of innervation (Fig 1A). Therefore, we sought to determine if synaptic losses are potentiated after noise exposure in the homozygous *Sirt3-KO* mouse. We imaged MYO7+ IHCs at 12 and 24 kHz that had also been stained with antibodies against CTBP2, the

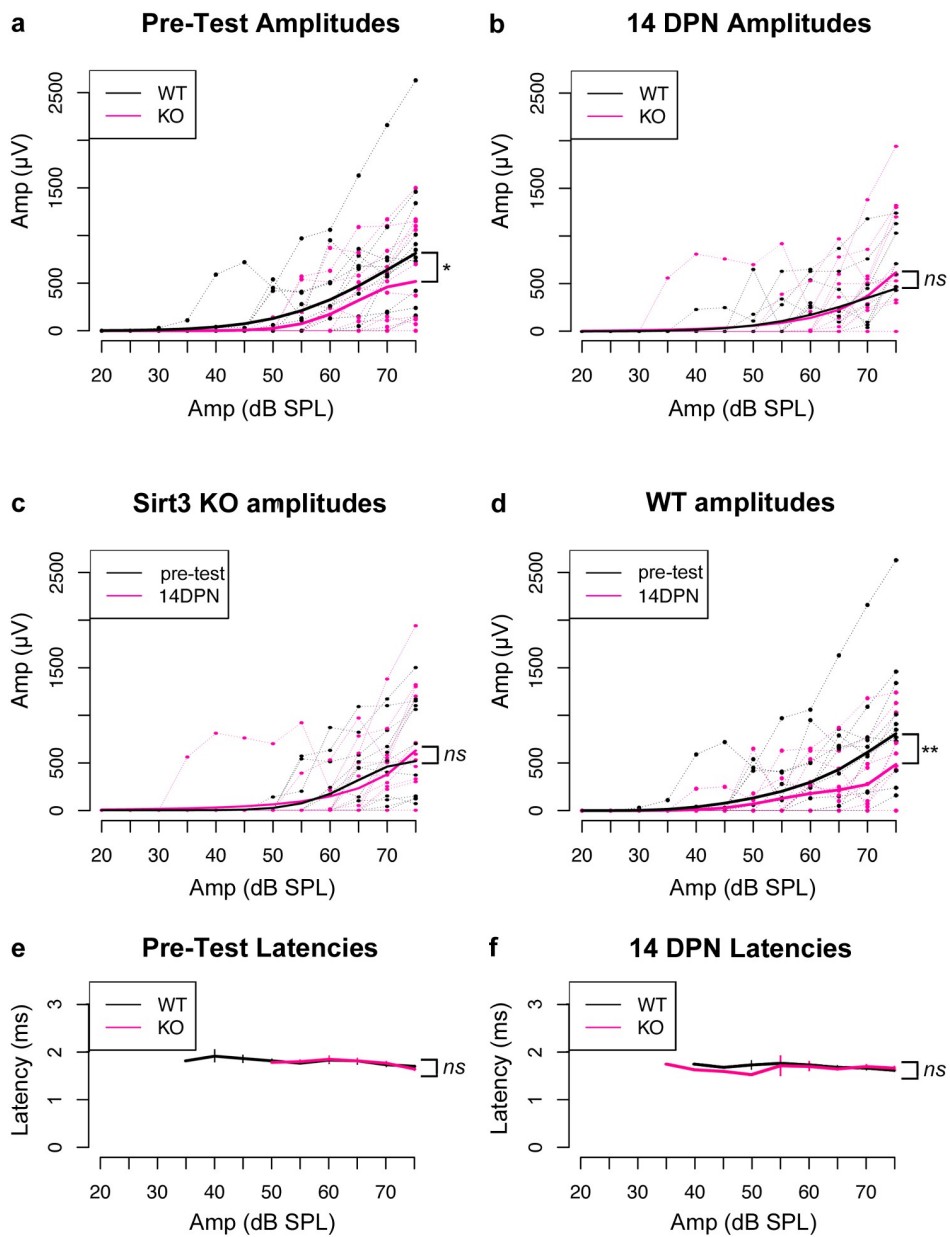

**Fig 3. SIRT3 loss of function impacts pre-noise ABR peak 1 amplitudes, but not amplitudes after noise exposure.**
a. Peak 1 amplitudes in ABR measurements for 32 kHz stimuli prior to noise exposure for homozygous *Sirt3-KO* mice
(n = 14, thin pink) and wild-type littermates (n = 13, thin black). Peak 1 amplitudes (microvolts) are plotted on the y
axis for different amplitudes of stimulus (dB SPL) plotted on the x-axis. Heavier pink and black lines represent
respective GEE models. The area under the curve, representative of the progressive neuronal recruitment, is
significantly reduced for homozygous *Sirt3-KO* mice (p = 0.046, parametric bootstrap method, n = 27 mice, indicated
with *). b. Same analysis as in (a), but for peak 1 amplitude values obtained from the same mice 14 DPN. No difference
is seen between genotypes (p = 0.87, parametric bootstrap method, n = 27 mice, indicated with *ns*). c. Same analysis as
in (a, b), however, in this case amplitudes obtained from homozygous *Sirt3-KO* mice are compared before and after
noise exposure. No differences are seen (p = 0.914, parametric bootstrap method, n = 14, indicated with *ns*). d. Same
analysis as in (a, b), however, in this case amplitudes obtained from wild-type littermates are compared before and
after noise exposure. A significant reduction in amplitude is evident after noise exposure (cf. red to black, p = 0.002,
parametric bootstrap method, n = 13, indicated with **). e. Mean peak 1 latencies for 32 kHz stimuli prior to noise
exposure for homozygous *Sirt3-KO* mice (n = 14, pink) and wild-type littermates (n = 13, black). No differences are
seen between genotypes (Kruskal-Wallis rank sum test, p = 0.68, indicated with *ns*). Error bars: s.e.m. f. Mean peak 1
latencies for 32 kHz stimuli at 14 DPN for homozygous *Sirt3-KO* mice (n = 14, pink) and wild-type littermates (n = 13,
black). No differences are seen between genotypes (Kruskal-Wallis rank sum test, p = 0.26, indicated with *ns*). Error
bars: s.e.m.

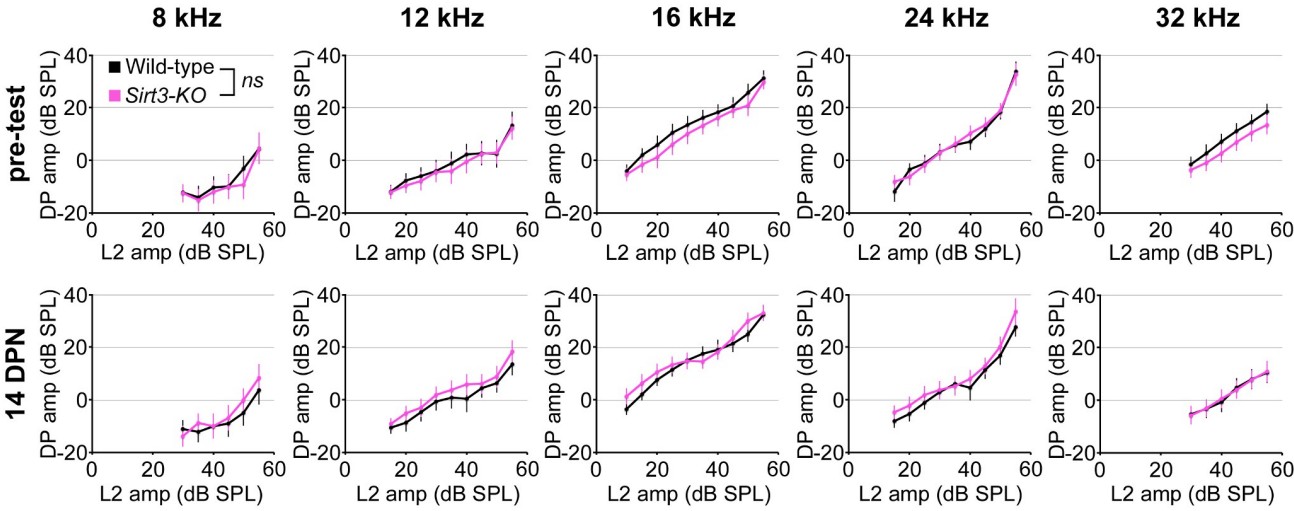

**Fig 4. SIRT3 loss of function does not impact OHC function as measured with DPOAE pre-noise input-output functions in either condition.** Ten graphs comparing homozygous *Sirt3-KO* (pink) and wild-type (black) littermates are arranged by time point, with pre-test results on the top row and 14 DPN results on the bottom row, as well as by frequency, displaying 8 kHz, 12 kHz, 16 kHz, 24 kHz, and 32 kHz results from left to right. Mean distortion product (DP) amplitudes in dB are plotted on the y-axis, and the amplitudes of the L2 stimulus (dB) are plotted on the y-axis. No differences between the genotypes (p = 0.07 for genotype, ANOVA, n = 27 mice, indicated with *ns*). Error bars: s.e.m.

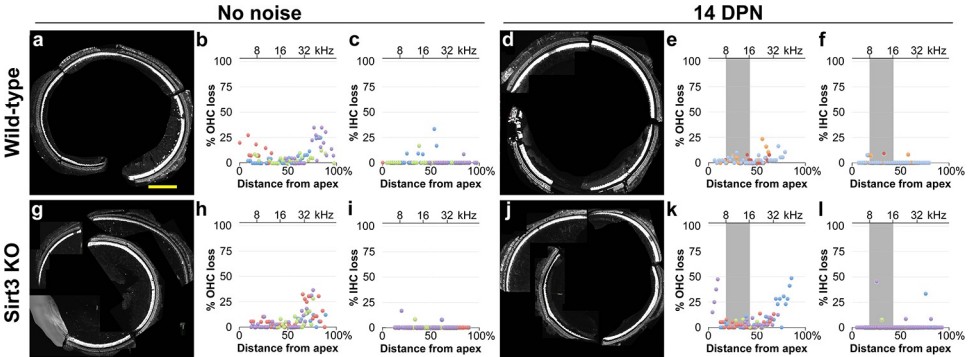

**Fig 5. SIRT3 loss of function does not impact OHC or IHC survival in either condition.** a. Representative cochlear preparation from a wild-type littermate without noise exposure, with IHCs revealed with anti MYO7 antibodies (white) and OHCs revealed with anti-OCM antibodies (also white). Scale bar: 200 microns. b. Cochleogram results from 4 mapped cochlear preparations from 4 wild-type littermates without noise exposure, where OHC loss was quantified in 100 micron segments. The distance from the apex for each segment is plotted on the x-axis, and the percent OHC loss for the segment is plotted on the y-axis. Dots with the same color (purple, blue, green, or red) are from the same cochlear preparation. c. Cochleogram results from the same preparations, where IHC loss was quantified in the same segments and presented similarly with corresponding colors. d. Representative cochlear preparation from a wild-type mouse at 14 DPN, with the same staining as in (a). e. Cochleogram results for OHCs, similar to (b), from 3 mapped cochlear preparations from 3 wild-type littermates at 14 DPN. Compared to (b), p = 0.18, Kruskal-Wallis rank sum test, n = 3,4 biological replicates. f. Cochleogram results for IHCs, similar to (c), from the same preparations shown in (d). Compared to (c), p = 0.80, Kruskal-Wallis rank sum test, n = 3,4 biological replicates. g. Representative cochlear preparation from a homozygous *Sirt3-KO* mouse without noise exposure, with the same staining as in (a). h. Cochleogram results for OHCs, similar to (b), from 4 mapped cochlear preparations from 4 homozygous *Sirt3-KO* mice without noise exposure. Compared to (b), p = 0.71, Kruskal-Wallis rank sum test, n = 4 biological replicates. i. Cochleogram results for IHCs, similar to (c), from the same preparations shown in (h). Compared to (c), p = 0.45, Kruskal-Wallis rank sum test, n = 4 biological replicates. j. Representative cochlear preparation from a homozygous *Sirt3-KO* mouse at 14 DPN, with the same staining as in (a). k. Cochleogram results for OHCs, from 4 mapped cochlear preparations from 4 homozygous *Sirt3-KO* mice at 14 DPN. Compared to (h), p = 0.49, Kruskal-Wallis rank sum test, n = 4 biological replicates. l. Cochleogram results for IHCs, similar to (c), from the preparations shown in (k). Compared to (i), p = 0.76, Kruskal-Wallis rank sum test, n = 4 biological replicates.

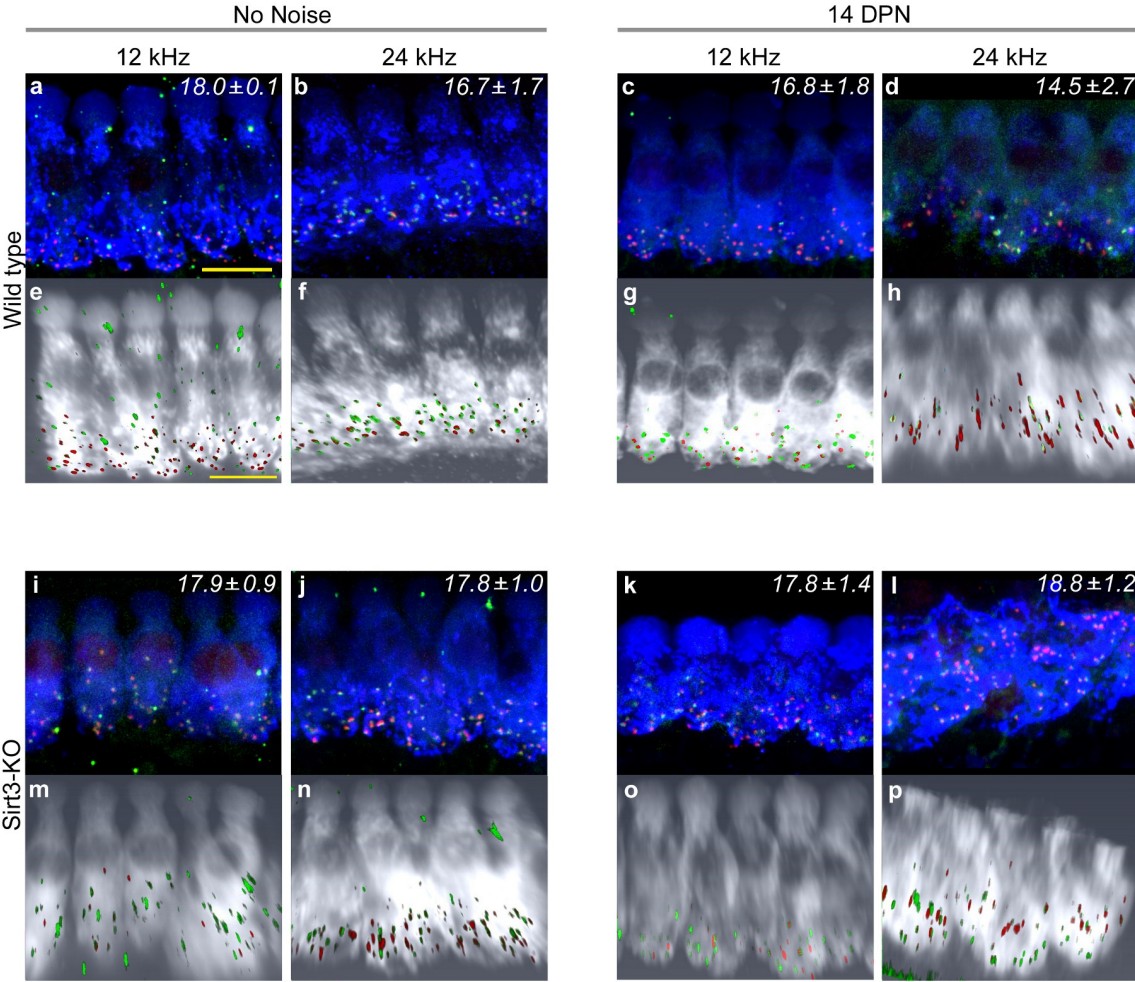

**Fig 6. SIRT3 loss of function does not increase synaptic loss in either condition.** a. Representative image of wild-type 12 kHz IHCs with no noise exposure, stained with anti-MYO7 (blue), anti-CTBP2 (red) to reveal pre-synaptic structures, and anti-GRIA2 (green) to reveal post-synaptic structures. Mean synaptic number per IHC (± s.e.m.), visually counted from 3 biological replicates with 8 IHCs each, is noted in the upper right hand corner. Size bar: 5 microns. b. Representative image of wild-type 24 kHz IHCs with no noise exposure, similarly stained to (a). 3 replicates, 6–7 IHCs each. c. Representative image of wild-type 12 kHz IHCs at 14 DPN, similarly stained to (a). 4 replicates, 7–11 IHCs each. d. Representative image of wild-type 24 kHz IHCs at 14 DPN, similarly stained to (a). 3 replicates, 7–8 IHCs each. e–h. Three dimensional Amira rendering of confocal stacks shown in (a–d), respectively, with MYO7 rendered in white, CTBP2 in red, and GRIA2 in green. Hair cells are rotated to display from similar orientations. i. Representative image of homozygous *Sirt3-KO* 12 kHz IHCs with no noise exposure, similarly stained to (a). 3 replicates, 6–8 IHCs each. j. Representative image of homozygous *Sirt3-KO* 24 kHz IHCs with no noise exposure, similarly stained to (a). 3 replicates, 4–7 IHCs each. k. Representative image of homozygous *Sirt3-KO* 12 kHz IHCs at 14 DPN, similarly stained to (a). The angle of the tissue in the imaging foreshortens the IHC. 5 replicates, 6–8 IHCs each. l. Representative image of homozygous *Sirt3-KO* 24 kHz IHCs at 14 DPN, similarly stained to (a). IHCs appear different because of the angle of the tissue. 3 replicates, 6–8 IHCs each. m—p. Three-dimensional Amira rendering of confocal stacks shown in (i–l), respectively, with MYO7 rendered in white, CTBP2 in red, and GRIA2 in green. Hair cells are rotated to display from similar orientations. The apical truncation shown in (p) is an artifact of the confocal imaging.

major component of auditory synaptic ribbons, as well as GRIA2, one of the glutamate receptor components of auditory post-synaptic specializations [33]. Representative images for wild-type and homozygous *Sirt3-KO* cochleae, with and without noise exposure, are depicted in Fig 6. Confocal stacks were imported into Amira, so that the IHCs could be rotated, to view at similar angles. Paired synaptic structures were quantified visually on randomized projected images, whose condition and genotype were masked from the analyst. Each frequency, condition, and genotype is labeled with the mean number of synapses per IHC for 3–4 biological

replicates. Wild-type cochlea harbored between 16.7 and 18.0 synapses per IHC in no noise condition (Fig 6a and 6b). Similar numbers were obtained after noise exposure (Fig 6c and 6d), as previously reported [21, 23]. At both frequencies and in both conditions, homozygous *Sirt3-KO* cochleae harbored between 17.8 and 18.8 synapses per IHC (Fig 6i–6l). As the numbers of auditory synapses in homozygous Sirt3-KO mice are the same or slightly more compared to wild-type littermates in either condition, we conclude that the loss of SIRT3 protein does not potentiate synaptopathy from noise exposure.

## Discussion

NIHL is a life-long, progressive disability that presents a significant social and personal burden. Variants in genes that reduce oxidative stress can enhance susceptibility to occupational NIHL [7], indicating the importance of this pathway to acquired hearing disorders. SIRT3 is a major regulator of the mitochondrial oxidative stress response [34], and the pharmacological activation of SIRT3 protects against permanent threshold shift (PTS) in mice [19]. We tested the hypothesis that loss of SIRT3 function impacts recovery from a noise insult that confers a temporary threshold shift (TTS) in young adult mice. We show that SIRT3 immunoreactivity is present in OHCs, around the post-synaptic structures of IHCs, and faintly within IHCs themselves, comparable to mRNA studies [30]. We measured auditory thresholds of homozygous *Sirt3*-KO mice and their wild-type littermates with ABR and DPOAE, before noise exposure, immediately after noise exposure, and a third time after the mice had recovered. We observed differences in auditory function between the genotypes prior to noise exposure. Homozygous Sirt3-KO mice initially had slightly, but significantly, worse overall ABR thresholds compared to their wild-type littermates. Moreover, homozygous Sirt3-KO mice initially had significantly reduced high-frequency ABR amplitude progressions compared to wild-type littermates. After noise exposure, no ABR differences were observed between the genotypes. We did not observe differences in OHC function, by DPOAE threshold or fine structure, or in OHC survival. No differences were observed in IHC survival in either condition. Homozygous *Sirt3-KO* mice had similar numbers of high-frequency IHC synapses to their wild-type littermates, both before and after noise exposure. These results suggest that SIRT3 may be important in spiral ganglion signaling, as indicated by the reduced ABR function prior to noise damage. They also indicate that SIRT3 is not required in mice for recovery from TTS, which contrasts with its clear role in mitigating PTS damage when exogenously activated.

The oxidative stress response acts to mitigate the effects of traumatic or repetitive noise injury on the cochlea [11]. Mitochondrial genetic defects are strongly associated with NIHL susceptibility [10], and strategies to reduce mitochondrial oxidative stress can protect from NIHL in animal models [35, 36]. SIRT3 protects by activating mitochondrial enzymes that reduce oxidative radicals, including SOD2 [37] and Catalase [38], through lysine deacetylation. In analyses of SIRT3 effector molecules, heterozygous *Sod2-KO* mice were shown to incur more damage from traumatic noise compared to wild-type controls, including higher ABR thresholds and greater OHC losses [39]. Notably, the exogenous activation of SIRT3 through pharmacological means restores partial cochlear function after traumatic noise damage [19] and aminoglycoside damage [40]. Taken together, these data show that in mice, the SIRT3-induced oxidative stress response is activated after traumatic noise, reduced levels of those effectors worsen trauma outcomes, and increasing SIRT3 activity improves trauma outcomes. This is consistent with current models of human occupational NIHL susceptibility, and informs some efforts to combat NIHL [41, 42].

In contrast, the experiments in this report assess hearing recovery from TTS. Studies investigating pharmacological activation of SIRT3 showed that loss of SIRT3 did not further

exacerbate damage from PTS in mice [19]. We have previously shown that the TTS paradigm reveals susceptibility to noise damage in homozygous *Foxo3-KO* mice [21]. Others have used TTS paradigms to investigate the protective roles of the medial olivocochlear system [43] and of peroxisomes [44]. In all three examples, a single sub-traumatic noise exposure readily revealed damage susceptibility. We find that there is no similar susceptibility with SIRT3 loss-of-function mice (Figs 2, 4 & 5), even though SIRT3 is detected in the organ of Corti (Fig 1) and impacts auditory function before noise exposure (Figs 2 & 3). These data may be interpreted to indicate that for a single, sub-traumatic noise exposure, protective mechanisms not related to oxidative stress are sufficient for hearing recovery. Cochlear responses to chronic insults, such as aging and occupational noise exposure, likely contain additional mechanisms. Nonetheless, a profound relationship between TTS and age-related hearing loss has been reported, in which acute synaptopathy was associated with an exacerbated long-term change of cochlear function [20]. In this regard, whether the presence of SIRT3 makes a difference in the situation of cochlear synaptopathy or in long-term changes after sub-traumatic noise exposure is worth further investigation.

We present a new analytical technique for calculating significant differences between amplitude progressions, which has two novel components. First, we compare the integrals of the amplitude progressions, as this measure reflects neuron recruitment in response to increasing stimuli. Since the summation of neuronal activity is the variable of interest, we contend that the integral of the amplitude progression function is the appropriate measure for comparison. This contrasts with comparing the slopes of two progressions, as these are the derivatives of the neuronal summation. Second, we use generalized estimating equations (GEE) instead of a linear extrapolation to model amplitude progressions [29]. GEE are commonly used in longitudinal biomedical and environmental studies to assess the effect of an intervention [45]. This is because GEE can account for the correlation structure that results from repeated measurements of the same individual, leading to more consistent estimation of differences when compared to parametric tests that neglect correlation [46]. Others have used GEE to assess differences over time in human retinal ERG outputs, which are similar in nature to ABR progressions [47, 48]. Of course, this method is only valid when thresholds are not statistically different, as we had found for the 32 kHz frequency (Fig 2).

We show that high-frequency cochlear synaptopathy from noise exposure is not potentiated in the homozygous *Sirt3-KO* mouse (Fig 6). This result was counterintuitive, given the role of excitotoxic glutamate in auditory synaptic loss [20, 49–51] and the presence of SIRT3 immunoreactivity around the synaptic region of the IHCs (Fig 1). The latter finding suggests that SIRT3 may promote mitochondrial function in neurites and supporting cells, and would be predicted to be protective. However, we note that the homozygous *Sirt3-KO* mouse would have had mitochondrial dysfunction throughout its development. It is possible that compensatory mechanisms to prevent calcium overload are present in homozygous *Sirt3-KO* neurites. Such mechanisms could take the form of changes in efferent neurotransmission. Exogenous dopamine, for example, can lower cochlear action potential amplitudes in guinea pigs, providing a tonic effect that counteracts excitotoxicity [52]. Alternatively, stress signaling within spiral ganglion afferent neurites could modulate their activity. Altered cAMP levels within spiral ganglion neurites, for example, modulate an inward potassium current through HCN channels, which shapes EPSP waveforms and modulates the probability of spike formation [53]. Compensatory mechanisms could also account for the reduced peak 1 amplitude observed in homozygous *Sirt3-KO* mice prior to noise exposure (Fig 3). Note that in the wild-type mice, peak 1 amplitudes are significantly reduced in response to noise [23] in the absence of synaptopathy (Fig 6), suggesting that this change could be due to reduced or desynchronized firing by spiral ganglion neurons.

Our data are most consistent with a role for SIRT3 in the maintenance of spiral ganglion function in the absence of noise exposure, and not in restoring auditory function from a TTS or hair cell survival. It is interesting that while exogenous SIRT3 activation can mitigate the effects of noise damage in mice, endogenous SIRT3 activity may not serve this function. This suggests that a comprehensive approach for addressing noise damage could include both an understanding cellular homeostasis as well as novel ways to augment its stability.

## Acknowledgments

We gratefully acknowledge Dr. Anne Luebke, who maintains the Small Animal Auditory Testing Core and Dr. V. Kaye Thomas, who maintains the Center for Advanced Light Microscopy and Nanoscopy, both at URMC. We also thank Ms. Holly Beaulac for a critical reading of the manuscript.

## Author Contributions

**Conceptualization:** Xiaodong Tan, Patricia M. White.

**Data curation:** Sally Patel, Lisa Shah, Natalie Dang.

**Formal analysis:** Sally Patel, Xiaodong Tan, Anthony Almudevar, Patricia M. White.

**Funding acquisition:** Patricia M. White.

**Investigation:** Sally Patel, Lisa Shah, Natalie Dang, Xiaodong Tan, Patricia M. White.

**Methodology:** Xiaodong Tan, Anthony Almudevar, Patricia M. White.

**Project administration:** Patricia M. White.

**Writing – original draft:** Xiaodong Tan, Anthony Almudevar, Patricia M. White.

**Writing – review & editing:** Xiaodong Tan, Patricia M. White.

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
