## [Decision Letter · Decision Letter 0]

18 Mar 2020

PONE-D-20-03576

Endogenous SIRT3 activity is dispensable for normal hearing recovery after noise exposure in young adult mice

PLOS ONE

Dear Dr. White,

Thank you for submitting your manuscript to PLOS ONE. After careful consideration, we feel that it has merit but does not fully meet PLOS ONE’s publication criteria as it currently stands. Therefore, we invite you to submit a revised version of the manuscript that addresses the points raised during the review process.

To enhance the reproducibility of your results, we recommend that if applicable you deposit your laboratory protocols in protocols.io, where a protocol can be assigned its own identifier (DOI) such that it can be cited independently in the future. For instructions see: http://journals.plos.org/plosone/s/submission-guidelines#loc-laboratory-protocols

We look forward to receiving your revised manuscript.

Kind regards,

Hela Azaiez, Ph.D

Academic Editor

PLOS ONE

Journal Requirements:

2. To comply with PLOS ONE submissions requirements, in your Methods section, please provide additional information on the animal research and ensure you have included details on (1) methods of sacrifice, (2) methods of anesthesia and/or analgesia, and (3) efforts to alleviate suffering.

Reviewers' comments:

Reviewer's Responses to Questions

**Comments to the Author**

1. Is the manuscript technically sound, and do the data support the conclusions?

Reviewer #1: Yes

Reviewer #2: Yes

2. Has the statistical analysis been performed appropriately and rigorously? 

Reviewer #1: I Don't Know

Reviewer #2: No

3. Have the authors made all data underlying the findings in their manuscript fully available?

Reviewer #1: Yes

Reviewer #2: Yes

4. Is the manuscript presented in an intelligible fashion and written in standard English?

Reviewer #1: Yes

Reviewer #2: Yes

5. Review Comments to the Author

Reviewer #1: The authors performed experiments of noise-induced hearing loss (temporary threshold shift) in Sirt3 knockout mice. The hearing levels (ABR and DPOAE) were assessed in Sirt3 KO mice and WT mice cochleae at the time points of pre-noise, 1 day post-noise (1DPN), and 14 days post-noise(14DPN). Synaptic counts of the inner hair cells, and cochleaograms were assessed at 1DPN and 14DPN.

The hearing levels of SIRT3 KO mice were not significantly different from WT mice at all time points by conventional ABR and DPOAE. The authors further analyzed ABR wave I amplitude by an advanced statitistical method, generalized estimating equations which showed significantly decreased wave 1 amplitude in the KO mice at the pre-noise time point.

Synaptic counts and cochleograms were not significantly different between Sirt3 KO mice and WT mice at pre-noise and 14DPN.

This reviewer think methods and data of this manuscript is scientifically solid, but these are basically negative data, in that Sirt3 activity is not necessary for spontaneous recovery from noise-induced hearing loss (temporary threshold shift) in mice.

For example in lines 422-424 it is discussed that “These data may be interpreted to indicate that for single, sub-traumatic noise exposure, some other cellular protective mechanism is sufficient for endogenous recovery. Heat shock proteins and the FOXO3-dependent response are candidates”, but these claims should be supported by an original data which directly address this discussion.

However this manuscript contains the below findings.

1) Wave 1 ABR amplitude was significantly reduced in Sirt3 knockout mice, as shown by generalized equation method.

2) Synaptic count in IHC was not different in Sirt3 knockout mice as compared to WT mice.

3) Therefore the reduced wave 1 amplitude may reflect importance of Sirt3 in the spiral neurons.

This reviewer recommend thorough revision of this manuscript (including the title) in line with the above positive findings in this data set. The below is additional comments:

1) Please indicate the reason why the authors focused on the function of Sirt3 in noise induced hearing loss of temporary threshold shift (TTS), instead of permanent threshold shift(PTS). What would be expected in cases of PTS?

2) Lines 452-453, “Generalized estimating equations can provide a function that is more accurate representation of the progression.”

Please indicate more solid basis supporting this clam, for example, by citing previous papers. This reviewer feel foundation of this methodology can be a topic of another separate research paper.

Reviewer #2: Several years ago, a group showed that pharmacologic activation of SIRT3, or genetic overexpression of SART3 had a protective effect on noise-induced hearing loss. This study was overall consistent with other studies which similarly did “gain of function” studies for SIRT3.

This study is interesting because it looks at the endogenous un-activated SIRT3. The data fairly convincingly shows that endogenous SART3 does not have a major role at its normal basal levels in protecting against noise-induced hearing loss. Characterization of the localization of SIRT3 is also done, and other useful information is acquired.

Overall, I felt the story as a whole was convincing, and except for statistical tests, I don’t think any significant work is needed. However, I think some textual revisions are warranted to really highlight the difference between the loss of function experiments performed here and the gain of function experiments performed elsewhere.

Suggestions:

1. I think the major suggestion would be to clarify the major hypothesis that is being tested in this entire paper within the abstract. The abstract is not convey the critical background information which is that gain of function and pharmacologic activation studies have been performed and they have demonstrated that gain of function activity has protective effects. The abstract needs to clarify this and also clarified that the outstanding question of whether the endogenous levels of SIRT3 have a role in the absence of activation. This is not well spelled out in the abstract, but it does come through in the introduction. Since many people will choose to read the paper based on the abstract, the abstract needs to be much clearer regarding the rationale for the current experiments. One could possibly read the abstract and think that the purpose of this group is to contradict the gain of function studies. That is definitely not the case, but the abstract is not clear about what is being tested and whether the purpose of this experiment is to reevaluate the gain of function studies, which certainly is not the goal. So, I suggest that the authors clarify that previous studies were gain of function studies and do not address the role of the endogenous protein in the absence of pharmacologic or genetic activation.

2. In line 322, and elsewhere, the authors say “strikingly, noise exposure did not potentiate OHC losses…” I don’t know why the authors say strikingly. To me, there’s no reason to think that endogenous levels of SIRT3, without activation, had any role. I would remove this type of language because it implies that the authors had a preconceived notion about the function of SIRT3, and a preconceived notion that the basal levels have a protective effect. I think they should present themselves as more objective throughout this manuscript making it clear that it is very plausible that endogenous SIRT3 has no beneficial effect versus the alternative model that endogenous SIRT3 has a beneficial effect. I think there’s no reason for the authors to take a stand one way or another so they should be clear in the introduction, first paragraph of the Discussion, and elsewhere that they had no bias and either result would be interesting and important.

3. In some places, the authors refer to “endogenous recovery.” I think this doesn’t make sense. All recovery is endogenous. I think they mean endogenous levels of SIRT3, or basal levels of SIRT3 activity or something else. But the term “endogenous recovery” is inaccurate and confusing. I would correct this throughout.

4. To me, the authors’ Discussion really miss is the most important and interesting part of the entire study which is that SIRT3 can be exploited for a novel function – protection from hearing loss, through pharmacologic or genetic activation. However, the protein normally does not serve this function. It only serves this function of highly activated through these gain of function approaches. They should be clearly stated and this is in fact very interesting. But this is the best and clearest way to explain the robust effects of gain of function work by reference 20 compared to the lack of effect with the loss of function experiments performed here. I would revise the discussion appropriately.

5. In many cases many of the figures black any statistical presentation. It’s hard to know what the key values are in several of the major figures such as figure 2, figure 3, and figure 4. Even if there are no statistically significant differences, these need to be indicated as well. The exact statistical tests need to be mentioned in the legends.

6. PLOS authors have the option to publish the peer review history of their article (what does this mean?). If published, this will include your full peer review and any attached files.

Reviewer #1: No

Reviewer #2: No

---

## [Author Response · Author response to Decision Letter 0]

28 May 2020

Response to reviewers for PONE-D-20-03576

Original title: Endogenous SIRT3 activity is dispensable for normal hearing recovery after noise exposure in young adult mice

New title: SIRT3 promotes auditory function in young adult FVB/nJ mice but is dispensable for hearing recovery after noise exposure 

Dear Dr. Azaiez,

We have submitted a revised form of our manuscript for review. We are grateful for the constructive comments by the reviewers and hope that this revision meets all of their requirements as well as PLOS One’s requirements. Please see our responses below.

On behalf of all the authors, I would like to thank the reviewers and the editor for their constructive advice on how to improve this work.

Journal Requirements: 

 We have brought the style of the manuscript into that required by PLOS ONE.

2. To comply with PLOS ONE submissions requirements, in your Methods section, please provide additional information on the animal research and ensure you have included details on (1) methods of sacrifice, (2) methods of anesthesia and/or analgesia, and (3) efforts to alleviate suffering.

 We have added details on the methods of euthanasia and tissue sampling, noted which procedures were performed without anesthesia, and described our efforts to assess and alleviate suffering of experimental subjects.

3. We note that you have included the phrase “data not shown” in your manuscript. Unfortunately, this does not meet our data sharing requirements. PLOS does not permit references to inaccessible data. 

 We require that authors provide all relevant data within the paper, Supporting Information files, or in an acceptable, public repository. Please add a citation to support this phrase or upload the data that corresponds with these findings to a stable repository (such as Figshare or Dryad) and provide and URLs, DOIs, or accession numbers that may be used to access these data. Or, if the data are not a core part of the research being presented in your study, we ask that you remove the phrase that refers to these data.

We have revised Figure 5 to include data on IHC survival, which had previously not been shown. We have searched the document and removed all phrases referring “data not shown” or “not shown”.

Reviewer #1: 

The authors performed experiments of noise-induced hearing loss (temporary threshold shift) in Sirt3 knockout mice. The hearing levels (ABR and DPOAE) were assessed in Sirt3 KO mice and WT mice cochleae at the time points of pre-noise, 1 day post-noise (1DPN), and 14 days post-noise(14DPN). Synaptic counts of the inner hair cells, and cochleaograms were assessed at 1DPN and 14DPN.

The hearing levels of SIRT3 KO mice were not significantly different from WT mice at all time points by conventional ABR and DPOAE. The authors further analyzed ABR wave I amplitude by an advanced statitistical method, generalized estimating equations which showed significantly decreased wave 1 amplitude in the KO mice at the pre-noise time point.

Synaptic counts and cochleograms were not significantly different between Sirt3 KO mice and WT mice at pre-noise and 14DPN.

This reviewer think methods and data of this manuscript is scientifically solid, but these are basically negative data, in that Sirt3 activity is not necessary for spontaneous recovery from noise-induced hearing loss (temporary threshold shift) in mice.

We thank the reviewer for their assessment of our scientific analysis. We agree that the TTS recovery data are essentially negative. We don’t consider this a problem, as negative results also contribute to the body of scientific knowledge.

For example in lines 422-424 it is discussed that “These data may be interpreted to indicate that for single, sub-traumatic noise exposure, some other cellular protective mechanism is sufficient for endogenous recovery. Heat shock proteins and the FOXO3-dependent response are candidates”, but these claims should be supported by an original data which directly address this discussion.

On the reviewer’s advice, we have removed our speculation about candidate gene pathways that might promote recovery in the absence of SIRT3.

However this manuscript contains the below findings.

1) Wave 1 ABR amplitude was significantly reduced in Sirt3 knockout mice, as shown by generalized equation method.

2) Synaptic count in IHC was not different in Sirt3 knockout mice as compared to WT mice.

3) Therefore the reduced wave 1 amplitude may reflect importance of Sirt3 in the spiral neurons.

This reviewer recommend thorough revision of this manuscript (including the title) in line with the above positive findings in this data set. The below is additional comments:

 We are grateful to the reviewer for their advice in interpreting our experiments. We have made revisions to the title, the abstract, and the discussion to highlight the positive results as recommended. We believe these include the validated antibody staining, which has not previously been published. In addition, the ABR tests before noise exposure indicated that Sirt3-KO mice had higher overall thresholds, even though no specific frequency was different. This was not emphasized in the original draft. Note, though, that our experimental design specifically addressed SIRT3’s role in recovery from sub-traumatic noise. We are not able to perform additional experiments on SIRT3’s role in spiral ganglion neurons, due to extraneous events, including our state and university’s shut down. We have highlighted the positive findings, while keeping the description of the experiments consistent with their design.

1) Please indicate the reason why the authors focused on the function of Sirt3 in noise induced hearing loss of temporary threshold shift (TTS), instead of permanent threshold shift(PTS). What would be expected in cases of PTS?

 Previous experiments performed by Brown et al, 2014 Cell Metab 20:1059-68 showed that SIRT3-KO and wild-type littermates had similar responses to PTS stimuli. We surmised that differences might be observed in response to TTS stimuli, as this approach has revealed roles for other proteins in cochlear homeostasis, as described in the Discussion (paragraph 3). We have revised the introduction and discussion to emphasize this point.

2) Lines 452-453, “Generalized estimating equations can provide a function that is more accurate representation of the progression.”

Please indicate more solid basis supporting this clam, for example, by citing previous papers. This reviewer feel foundation of this methodology can be a topic of another separate research paper.

 We regret that our justification for the use of generalized estimating equations was unclear. Our innovation is two-fold. First, we compare the integrals of the amplitude progressions, as these reflect the coordinated signaling of increasing numbers of spiral ganglion neurons with increasing stimulus. Second, we use GEE to model the progressions and thus to obtain the integrals. GEE are commonly used in longitudinal biomedical and environmental studies to assess the effect of an intervention (Liu and Colditz, 2018, Biom J 59:315-330). This is because GEE can account for the correlation structure that results from repeated measurements of the same individual, leading to more consistent estimation of differences when compared to parametric tests that neglect correlation Nitta et al, J. Epidemiol 2010, 20:117-184). Others have used GEE to assess differences in retinal ERG outputs in disease, which are similar in nature to ABR progressions (see Francis et al, 2014, PLOS ONE 9:e84247 and Pearl et al 2017 J. Huntingtons Dis 6:237-247 for examples). 

 Our use of GEE in assessing changes in the integrals of ABR progressions in subjects after noise exposure is novel and there are no previous citations on this application. Our choice of statistical analysis arose from consultations between Dr. White and Dr. Almudevar, who has decades of experience in biostatistical analyses. We have included this language and these references in the discussion section that describes the GEE, providing a clearer justification for its use.

Reviewer #2: Several years ago, a group showed that pharmacologic activation of SIRT3, or genetic overexpression of SART3 had a protective effect on noise-induced hearing loss. This study was overall consistent with other studies which similarly did “gain of function” studies for SIRT3.

This study is interesting because it looks at the endogenous un-activated SIRT3. The data fairly convincingly shows that endogenous SART3 does not have a major role at its normal basal levels in protecting against noise-induced hearing loss. Characterization of the localization of SIRT3 is also done, and other useful information is acquired.

Overall, I felt the story as a whole was convincing, and except for statistical tests, I don’t think any significant work is needed. However, I think some textual revisions are warranted to really highlight the difference between the loss of function experiments performed here and the gain of function experiments performed elsewhere.

 We thank the reviewer for their kind words about our work. We have revised the text as you suggested below.

Suggestions:

1. I think the major suggestion would be to clarify the major hypothesis that is being tested in this entire paper within the abstract. The abstract is not convey the critical background information which is that gain of function and pharmacologic activation studies have been performed and they have demonstrated that gain of function activity has protective effects. The abstract needs to clarify this and also clarified that the outstanding question of whether the endogenous levels of SIRT3 have a role in the absence of activation. This is not well spelled out in the abstract, but it does come through in the introduction. Since many people will choose to read the paper based on the abstract, the abstract needs to be much clearer regarding the rationale for the current experiments. One could possibly read the abstract and think that the purpose of this group is to contradict the gain of function studies. That is definitely not the case, but the abstract is not clear about what is being tested and whether the purpose of this experiment is to reevaluate the gain of function studies, which certainly is not the goal. So, I suggest that the authors clarify that previous studies were gain of function studies and do not address the role of the endogenous protein in the absence of pharmacologic or genetic activation.

 We have revised the abstract as recommended, including the hypothesis (“Here we test the hypothesis that SIRT3 is required in mice for recovery of auditory thresholds after a noise exposure that confers a temporary threshold shift.“) We also include a sentence to better place our results in the context of previous findings (“These loss-of-function studies contrast with previously published gain-of-function studies and help refine our understanding of SIRT3’s role in homeostasis.”)

2. In line 322, and elsewhere, the authors say “strikingly, noise exposure did not potentiate OHC losses…” I don’t know why the authors say strikingly. To me, there’s no reason to think that endogenous levels of SIRT3, without activation, had any role. I would remove this type of language because it implies that the authors had a preconceived notion about the function of SIRT3, and a preconceived notion that the basal levels have a protective effect. I think they should present themselves as more objective throughout this manuscript making it clear that it is very plausible that endogenous SIRT3 has no beneficial effect versus the alternative model that endogenous SIRT3 has a beneficial effect. I think there’s no reason for the authors to take a stand one way or another so they should be clear in the introduction, first paragraph of the Discussion, and elsewhere that they had no bias and either result would be interesting and important.

 We have revised the introduction, the results, and the discussion to present the work with less experimenter bias. We continue to describe the lack of synaptopathy potentiation as counter-intuitive, however, as it allows us to speculate on potential compensatory mechanisms.

3. In some places, the authors refer to “endogenous recovery.” I think this doesn’t make sense. All recovery is endogenous. I think they mean endogenous levels of SIRT3, or basal levels of SIRT3 activity or something else. But the term “endogenous recovery” is inaccurate and confusing. I would correct this throughout.

 We thank the reviewer for their advice, and have replaced “endogenous recovery” with “hearing recovery” throughout the manuscript.

4. To me, the authors’ Discussion really miss is the most important and interesting part of the entire study which is that SIRT3 can be exploited for a novel function – protection from hearing loss, through pharmacologic or genetic activation. However, the protein normally does not serve this function. It only serves this function of highly activated through these gain of function approaches. They should be clearly stated and this is in fact very interesting. But this is the best and clearest way to explain the robust effects of gain of function work by reference 20 compared to the lack of effect with the loss of function experiments performed here. I would revise the discussion appropriately.

 We now discuss this observation to the final paragraph of the discussion. 

5. In many cases many of the figures black any statistical presentation. It’s hard to know what the key values are in several of the major figures such as figure 2, figure 3, and figure 4. Even if there are no statistically significant differences, these need to be indicated as well. The exact statistical tests need to be mentioned in the legends.

 In figures where data are directly compared within a graph such as ABR thresholds, we have labeled that graph with the statistical presentation. In the DPOAE input-output functions, the overall statistical test showed no difference, and we have made a note of that one time on the figure. In the cochleograms, the data for each genotype and condition are presented separately, and we have noted in the legends which panels are compared and what their statistical tests are. For synaptic numbers, on inspection homozygous Sirt3-KO mice have equal numbers or on average slightly more synapses, which disproves the hypothesis that loss of Sirt3 confers sensitivity to synaptic loss from noise.

---

## [Decision Letter · Decision Letter 1]

17 Jun 2020

SIRT3 promotes auditory function in young adult FVB/nJ mice but is dispensable for hearing recovery after noise exposure

PONE-D-20-03576R1

Dear Dr. White,

We’re pleased to inform you that your manuscript has been judged scientifically suitable for publication and will be formally accepted for publication once it meets all outstanding technical requirements.

Kind regards,

Hela Azaiez, Ph.D

Academic Editor

PLOS ONE

Additional Editor Comments (optional):

Reviewers' comments:

Reviewer's Responses to Questions

**Comments to the Author**

1. If the authors have adequately addressed your comments raised in a previous round of review and you feel that this manuscript is now acceptable for publication, you may indicate that here to bypass the “Comments to the Author” section, enter your conflict of interest statement in the “Confidential to Editor” section, and submit your "Accept" recommendation.

Reviewer #1: (No Response)

Reviewer #2: All comments have been addressed

2. Is the manuscript technically sound, and do the data support the conclusions?

Reviewer #1: Yes

Reviewer #2: Yes

3. Has the statistical analysis been performed appropriately and rigorously? 

Reviewer #1: N/A

Reviewer #2: Yes

4. Have the authors made all data underlying the findings in their manuscript fully available?

Reviewer #1: Yes

Reviewer #2: Yes

5. Is the manuscript presented in an intelligible fashion and written in standard English?

Reviewer #1: Yes

Reviewer #2: Yes

6. Review Comments to the Author

Reviewer #1: Finally, please add the protocol number and the principle investigator name for the approval from the local animal research committee.

Reviewer #2: The manuscript is improved in terms of clarity, figure presentation, and statistical issues. I have no further concerns or suggestions.

7. PLOS authors have the option to publish the peer review history of their article (what does this mean?). If published, this will include your full peer review and any attached files.

Reviewer #1: No

Reviewer #2: No

---

## [Editor Report · Acceptance letter]

29 Jun 2020

PONE-D-20-03576R1 

SIRT3 promotes auditory function in young adult FVB/nJ mice but is dispensable for hearing recovery after noise exposure 

Dear Dr. White:

I'm pleased to inform you that your manuscript has been deemed suitable for publication in PLOS ONE. Congratulations! Your manuscript is now with our production department. 

Kind regards, 

on behalf of

Dr. Hela Azaiez 

Academic Editor

PLOS ONE